# SLogic: Subgraph-Informed Logical Rule Learning for Knowledge Graph Completion

## Abstract

Logical rule-based methods offer an interpretable approach to knowledge graph completion (KGC) by capturing compositional relationships in the form of human-readable inference rules. While existing logical rule-based methods learn rule confidence scores, they typically assign a *global* weight to each rule schema, applied uniformly across the graph. This is a significant limitation, as a rule's importance often varies depending on the specific query instance. To address this, we introduce **SLogic** (Subgraph-Informed Logical Rule learning), a novel framework that assigns *query-dependent* scores to logical rules. The core of SLogic is a context-aware scoring function. This function determines the importance of a rule by analyzing the subgraph locally defined by the query's head entity, thereby enabling a differentiated weighting of rules specific to their local query contexts. Extensive experiments on benchmark datasets show that SLogic outperforms existing rule-based methods and achieves competitive performance against state-of-the-art baselines. It also generates query-dependent, human-readable logical rules that serve as explicit explanations for its inferences.

## 1 Introduction

Knowledge graphs (KGs) capture complex relationships among diverse real-world entities, including commonly used ones such as people, locations, organizations, events, products, and concepts, as well as more specialized types such as genes, diseases, chemicals, publications, and technical terms. They are foundational to many applications, such as recommendation systems and question answering Hildebrandt et al. (2019); Lan & Jiang (2020).

Despite their usefulness, KGs are notoriously incomplete. To address this, the field of Knowledge Graph Completion (KGC) has explored several dominant paradigms. Embedding-based models (e.g., (Bordes et al., 2013; Yang et al., 2014; Dettmers et al., 2018; Trouillon et al., 2016; Sun et al., 2019)) learn vector representations of entities and relations to predict missing links. More recently, Graph Neural Networks (GNNs) based models (Zhang & Yao, 2022; Zhu et al., 2021; 2023) have achieved state-of-the-art performance by capturing intricate topological patterns within the graph's structure. However, the strength of both these approaches is also their primary weakness: their reasoning is opaque. They operate on dense, sub-symbolic vectors, they function as "black box" models, making it difficult to understand or trust their predictions in critical applications.

As an alternative, rule-based reasoning offers a transparent and explainable approach to KGC. These methods infer missing links by discovering and applying logical rules (e.g., $bornIn(X,Y) \land locatedIn(Y,Z) \rightarrow livesIn(X,Z)$), which are inherently human-readable. This interpretability is a significant advantage, providing not just a prediction, but also the logical path to reach it. While rule-based models offer inductive capabilities and can outperform many embedding-based methods, a common limitation is that they learn one weight for each rule across the entire graph, despite the fact that weight learning takes into account the grounding context (e.g., (Yang et al., 2017; Sadeghian et al., 2019)). This overlooks the fact that a rule's relevance can change depending on the specific query instance (e.g., a rule may be 80% confident for one query but only 50% for another), and therefore, answers to a query should be *context-dependent*.

For a concrete example, consider the KGC query $livesIn(Person, ?)$. One plausible rule, $bornIn(X,Y) \land locatedIn(Y,Z) \rightarrow livesIn(X,Z)$, might link a person to their birth location. A second rule, $worksAt(X,Y) \land locatedIn(Y,Z) \rightarrow livesIn(X,Z)$, could link them to their place

of work. A model relying on global rule confidences, perhaps favoring the globally more common "birthplace" rule, would fail to distinguish the second rule is more relevant in the context of a CEO of a major tech company because the "$worksAt$" path becomes exceptionally reliable. Our approach uses query-specific subgraphs to dynamically assess the relevance of each potential reasoning path for a given query, capturing the nuance that global scoring models miss.

To capture and utilize the local importance of a rule, we propose a novel hybrid approach, **SLogic** (Subgraph-Informed Logical Rule learning), to enhance rule-based reasoning with contextual awareness. SLogic pre-calculates simple paths to serve as then explicit, interpretable rule base. It then employs a GNN not as the final predictor, but as a powerful context encoder for the query. The contribution of SLogic is twofold. (i) It learns query-dependent weights that improve prediction accuracy, particularly on datasets where global rules are ambiguous. (ii) The learned scores provide granular insight into the reasoning process, identifying not just which rules are invoked, but how much each contributes to completing a specific edge. To the best of our knowledge, SLogic is the first neuro-symbolic framework to move beyond learning global rule confidence, introducing a mechanism to learn rule weights tailored to specific query contexts. We emphasize that while SLogic outperforms other rule-based approaches, it is best understood as an interpretable rule-learning framework rather than a purely metric-driven link predictor.

The rest of the paper is organized as follows. Section 2 briefly presents a literature review. Section 3 introduces the terminology, notations, and the research problem. Section 4 explains our proposed method. Section 5 presents the experimental results and Section 6 concludes the work.

## 2 Related Work

The literature on knowledge graph completion (KGC) can be broadly categorized into embedding-based approaches (Bordes et al., 2013; Yang et al., 2014; Dettmers et al., 2018; Trouillon et al., 2016; Sun et al., 2019), GNN-based methods (Zhang & Yao, 2022; Zhu et al., 2021; 2023), and rule-based strategies. Among GNN-based methods, NBFNet (Zhu et al., 2021) has achieved state-of-the-art performance by capturing all-path information through a neural Bellman-Ford message passing mechanism. While such methods are highly effective, they generally either operate as black boxes or provide path weights without learning explicit, human-readable rules that rule-based approaches offer. As explained in Section 1, rule-based reasoning enhances the explainability of knowledge graph completion. We therefore focus on describing such methods.

Existing rule-based methods can be broadly classified according to how they discover and learn logical rules. Probabilistic rule learning frameworks, such as Markov Logic Networks (MLNs) (Richardson & Domingos, 2006) and pLogicNet (Qu & Tang, 2019), combine logic with probabilistic models to learn weights for logical rules. A different line of work focuses on learning rule confidences by actively searching for paths (groundings) on the graph during the training process. NeuralLP (Yang et al., 2017) and its successor DRUM (Sadeghian et al., 2019) use a differentiable framework inspired by TensorLog (Cohen et al., 2017) to find soft proofs. Similarly, RNN-Logic (Qu et al., 2021) is a probabilistic model that treats logic rules as a latent variable and uses an Expectation-Maximization (EM) algorithm to simultaneously train a rule generator and a reasoning predictor. Recent approaches such as RLogic (Cheng et al., 2022) and NCRL (Cheng et al., 2023) learn a neural function to score the quality of potential rule structures. They perform an exhaustive, brute-force enumeration of all possible rules up to a certain length. Another line of research explores hybrid frameworks that jointly learn embeddings and logical rules. Methods such as IterE (Zhang et al., 2019) and RPJE (Niu et al., 2020) iteratively update both components, where learned rules help refine entity and relation embeddings, and embeddings in turn guide the rule mining process. These prior efforts learn or rely on rules deemed important at the level of the entire knowledge graph. In contrast, our SLogic framework recalculates rule importance with respect to contextual entities and their surrounding subgraphs.

## 3 Preliminaries: Knowledge Graphs, Rules, and Completion

**Knowledge Graphs, Rules.** A knowledge graph (KG) is a structured representation of factual information, formally defined as a directed, multi-relational graph $\mathcal{G} = (\mathcal{E}, \mathcal{R}, \mathcal{T})$. Here, $\mathcal{E}$ is a finite set of entities (nodes) and $\mathcal{R}$ is a finite set of relations (edge types). The graph's structure is

composed of a set of factual triples $\mathcal{T} \subseteq \mathcal{E} \times \mathcal{R} \times \mathcal{E}$. Each triple $(h, r, t)$ represents a known fact, where $h \in \mathcal{E}$ is a head entity, $t \in \mathcal{E}$ is a tail entity, and $r \in \mathcal{R}$ is the relation that connects them.

A **path** from a node $v_i$ to a node $v_j$ is defined as a sequence $v_i \xrightarrow{r_1} v_{i_1} \cdots \xrightarrow{r_m} v_j$, where each edge is labeled by a relation. **A path is simple** if it does not contain repeated nodes. The corresponding **relational path** is the ordered list of relations along the path, $(r_1, \ldots, r_m)$.

Our proposed method is based on logical rules. A **logical rule** is defined as a Horn clause, where the head is a single atomic formula (the conclusion) and the body is a conjunction of atomic formulas (the premises), i.e., each rule has the form: $r_h(X, Y) \leftarrow r_1(X, Z_1) \wedge r_2(Z_1, Z_2) \wedge \cdots \wedge r_L(Z_{L-1}, Y)$. Here, the rule asserts that the target relation $r_h$ is likely to hold between entities $X$ and $Y$ if there exists an intermediate path connected by a sequence of relations, where the maximum body length $L$ is a predefined hyperparameter. For conciseness, we represent the relational path in the rule body as a single vector, $\mathbf{r}_b = (r_1, r_2, \ldots, r_L)$, which allows us to simplify the rule notation to $r_h \leftarrow \mathbf{r}_b$.

**A rule $r_h \leftarrow \mathbf{r}_b$ is locally applicable w.r.t. an entity** $h$, if its body path $\mathbf{r}_b$ can be successfully grounded starting from the head entity $h$. A **hard negative rule w.r.t. a triplet** $(h, r, t)$ is defined to be a rule that is both *globally high-quality* (high static confidence) and *locally applicable*.

This paper addresses the knowledge graph completion (KGC) problem. Given a knowledge graph $\mathcal{G} = (\mathcal{E}, \mathcal{R}, \mathcal{T})$ and a query $\mathbf{q} = (h, r, ?)$, the task is to identify the most plausible answer entities.

# 4 THE SLOGIC FRAMEWORK

Unlike traditional methods that assign static confidence scores to logical rules, SLogic learns a dynamic, query-dependent scoring function $\phi(h, r, \mathbf{r}_b)$. This function assesses the relevance of a candidate rule $r \leftarrow \mathbf{r}_b$ by considering not only the rule itself but also the rich structural context of the query's head entity, $h$. In order to calculate a context-aware score for a relation $r$, we design a novel strategy to generate instances that are enriched by context-aware rules and to train a model.

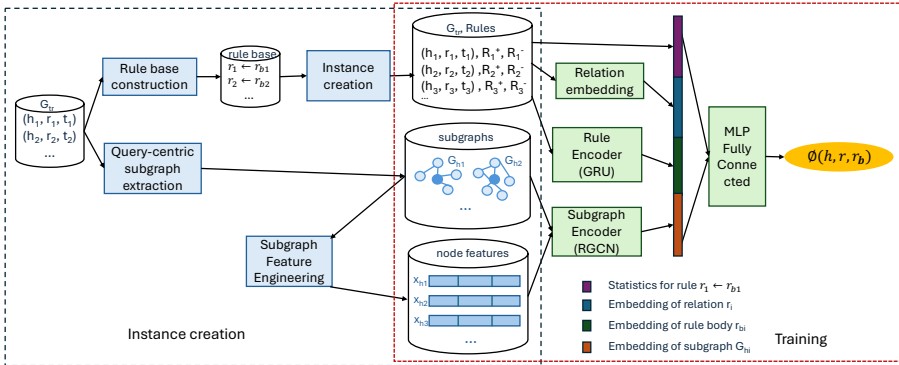

Figure 1: Steps to generate instances enriched by context aware rules and to train SLogic model

Figure 1 shows the major steps to create the rule enriched instances for a knowledge graph $\mathcal{G}_{tr}$ and to train the SLogic model. Sections 4.1 and 4.2 presents these components in detail.

## 4.1 GENERATION OF INSTANCES ENRICHED BY CONTEXT AWARE RULES

One novel component of this work is to utilize the context-aware rules. We do this by constructing instances that are enriched by context-aware rules.

### 4.1.1 RULE BASE CONSTRUCTION

The initial step of our method is to mine a comprehensive set of logical rules from the training knowledge graph $\mathcal{G}$. To extract these rules, we iterate through each ground truth triple $(h, r_h, t)$ in the training set. For each triple, we treat it as a positive example of a rule instantiation where $r_h$ is the head relation. We then perform a Depth-First Search (DFS) to find all simple paths connecting the

head entity $h$ and the tail entity $t$ within the graph, up to the predefined maximum length $L$. Each discovered path is converted into its corresponding relational path, which forms a candidate rule body $\mathbf{r}_b$. After enumerating all candidate rules, we quantify their quality using established statistical metrics. For each rule $r_h \leftarrow \mathbf{r}_b$, we compute its *standard confidence*, defined as the conditional probability $P(r_h|\mathbf{r}_b)$,

$$\text{Confidence}(r_h \leftarrow \mathbf{r}_b) = \frac{\#(\mathbf{r}_b, r_h)}{\#(\mathbf{r}_b)}$$

where $\#(\mathbf{r}_b, r_h)$ is the count of entity pairs connected by both the body path $\mathbf{r}_b$ and the head relation $r_h$, and $\#(\mathbf{r}_b)$ is the total count of entity pairs connected by the body path.

While standard confidence is widely used, it can be unreliable for rules where the body count $\#(\mathbf{r}_b)$ is low. To address this, we further compute the *Wilson score interval's lower bound* (Wilson, 1927) for each rule's confidence. This provides a more robust and conservative estimate of a rule's reliability, particularly for less frequent patterns. This score is calculated as

$$\text{Wilson}(p, n, z) = \frac{1}{1 + \frac{z^2}{n}} \left( p + \frac{z^2}{2n} - z\sqrt{\frac{p(1-p)}{n} + \frac{z^2}{4n^2}} \right)$$

where $p$ is the observed standard confidence, $n$ is the body count $\#(\mathbf{r}_b)$, and $z$ is the quantile of the standard normal distribution (typically 1.96 for a 95% confidence interval).

This mining process results in a static global rule base where each rule is associated with a confidence and a Wilson score. This base forms the symbolic foundation that SLogic later enriches with query-specific, contextual information.

### 4.1.2 QUERY-CENTRIC SUBGRAPH EXTRACTION

To provide a localized, computationally tractable context for each query, we perform an offline preprocessing step to extract a unique subgraph for every entity in the knowledge graph. This process yields a collection of graph structures that serve as the primary input to the subgraph encoder in SLogic (Figure 1).

To extract subgraphs for each entity $h \in \mathcal{E}$, we extract its local neighborhood by initiating a $k$-hop Breadth-First Search (BFS) in $\mathcal{G}$. This traversal expands outwards from the central entity $h$, exploring both its incoming and outgoing connections to capture a rich, bidirectional context. To maintain a uniform structure and mitigate the computational challenges posed by high-degree "hub" nodes, we employ neighbor sampling at each hop of the BFS. Specifically, if an entity has more neighbors than a predefined threshold ($\alpha$), we randomly pick $\alpha$ neighbors to continue the traversal. This results in a subgraph $\mathcal{G}_h \subset \mathcal{G}$ for entity $h$, which captures $h$'s most relevant local structure.

**Feature engineering.** A raw subgraph is insufficient for a GNN to distinguish node roles. We therefore enrich the subgraph with node features $\mathbf{x}_v$. Following SEAL (Zhang & Chen, 2018) and GraIL (Teru et al., 2020), we employ relative node labeling to capture structural patterns independent of node identities. The feature vector $\mathbf{x}_v$ is a concatenation of three components encoding the node's structural role relative to the query entity $h$.

(i) *Head Entity Indicator:* A two-dimensional binary vector that identifies whether a node is the query head entity $h$ (i.e., $[1, 0]$) or a neighbor (i.e., $[0, 1]$). This anchors the subgraph representation around the query. (ii) *Shortest Path Distance:* The geodesic distance (i.e., minimum number of hops) from node $v$ to the head entity $h$ within the subgraph $\mathcal{G}_h$. This explicitly encodes the structural proximity of each node to the query's origin. (iii) *Global Centrality Score:* The log-scaled degree of node $v$ as calculated from the complete graph $\mathcal{G}$. This feature injects a measure of the node's global importance into the local subgraph context.

These features are purely topological and entirely independent of any node-specific content or identifiers (e.g., entity IDs or pre-trained embeddings). *This design choice enforces the inductive capability of our model*, allowing it to generate meaningful representations for entities and subgraphs not encountered during training.

### 4.1.3 INSTANCE CREATION AND NEGATIVE SAMPLING

To build the training set from a knowledge graph $\mathcal{G}_{tr}$, for each ground truth triple $(h, r, t) \in \mathcal{G}_{tr}$, we first sample up to $k_{pos}$ unique positive rule bodies that correctly derive $t$. Then, for each of

these positive rules, we pair it with $k_{neg}$ hard negative rules selected using our newly designed sampling strategy described below. This creates a rich set of up to $k_{pos} \times k_{neg}$ training pairs for each original fact $(h, r, t)$. The parameters $k_{pos}$ and $k_{neg}$ are important hyperparameters that control the data diversity and the positive-to-negative ratio in our training objective. Their impact on model performance is thoroughly examined in our analysis (Section 5.2).

For each fact, besides providing the positive rules, we also generate hard negative rules motivated by the use of negative samples in their training process Mikolov et al. (2013). The use of such hard negative rules is to distinguish between correct and incorrect patterns and improves generalization. The selection process of hard negative rules involves two main stages. First, for a given query $(h, r, ?)$, we form a candidate pool by identifying all locally applicable rules for relation $r$ that can be successfully grounded from $h$, ranking them by their static Wilson score, and selecting the top-$K$. Next, from this pool, we remove the "positive" rules that lead to the true answer $t$ and then randomly sample $k_{neg}$ rules from the remaining set to serve as our hard negatives.

For a query $(h_i, r_i, t_i)$, let $\mathbf{R}_i^+$ and $\mathbf{R}_i^-$ consist of the $k_{pos}$ positive rules $\{r_i \leftarrow r_{b_{i1}}^+, \cdots, r_i \leftarrow r_{b_{ik_{pos}}}^+\}$ and $k_{neg}$ hard negative rules $\{r_i \leftarrow r_{b_{i1}}^-, \cdots, r_i^- \leftarrow r_{b_{ik_{neg}}}^-\}$ respectively. The instances generated in this step, subsequently provided to our model, comprise rule-enriched triplets, $((h_1, r_1, t_1), \mathbf{R}_1^+, \mathbf{R}_1^-), ((h_2, r_2, t_2), \mathbf{R}_2^+, \mathbf{R}_2^-)$, etc.

## 4.2 MODEL ARCHITECTURE, LOSS FUNCTION, AND TRAINING

The SLogic framework is composed of two primary neural encoders ((see training box of Figure 1).) The first one is a *subgraph encoder*, which uses a Relational Graph Convolutional Network (R-GCN) (Schlichtkrull et al., 2018) to process the query-centric subgraph $\mathcal{G}_h$. The second one is a *rule encoder*, which employs a Gated Recurrent Unit (GRU) (Cho et al., 2014) to encode the sequential rule body $\mathbf{r}_b$.

The embeddings from these two encoders, along with an embedding of the query relation $r$ and the pre-computed static features of the rule (e.g., confidence, support), are concatenated and passed through a final Multi-Layer Perceptron (MLP) to yield the query-specific rule score $\phi(h, r, \mathbf{r}_b)$.

We train SLogic using a learning-to-rank framework. The objective is to assign a higher score to a "positive" rule that correctly entails a known fact than to a "negative" rule that does not. We employ a margin-based ranking loss for each training pair.

$$\mathcal{L} = \max(0, \epsilon - (\phi(h, r, \mathbf{r}_b^+) - \phi(h, r, \mathbf{r}_b^-))) \tag{1}$$

where $\epsilon$ is a predefined margin hyperparameter, $\mathbf{r}_b^+$ is the body of a positive rule, and $\mathbf{r}_b^-$ is the body of a hard negative rule.

The model learns a scoring function $\phi(\cdot)$ for each rule $r \leftarrow \mathbf{r}_b$ w.r.t. a query $(h, r, ?)$ (i.e., $\phi(h, r, \mathbf{r}_b)$). This strategy forces the model to leverage the contextual information in the query-specific subgraph $\mathcal{G}_h$ to distinguish between globally plausible rules and those that are truly relevant to a possible query $(h, r, ?)$.

## 4.3 INFERENCE AND FINAL RANKING

At inference time, our goal is to answer a query $\mathbf{q} = (h, r, ?)$ by leveraging our trained SLogic model to find the most plausible reasoning paths. Our method operates as a re-ranking framework, first identifying a set of high-quality candidate rules and then using SLogic to score them based on the specific query context. This is followed by a mathematically-grounded aggregation step to produce the final answer ranking.

Figure 2 shows the steps to answer the given query. Step 1 generates candidate rules. For any given query $(h, r, ?)$, exhaustively scoring all rules in the global rule base would be computationally prohibitive. To obtain a manageable yet high-quality candidate set, we first identify all rules that are locally applicable to $h$, and then select the top-$N$ among them based on their Wilson score.

Step 2 calculates a context-aware (localized) importance score for each candidate rule. In particular, each candidate rule $\mathbf{r}_{b_i}$ is passed to our trained scoring function $\phi(h, r, \mathbf{r}_{b_i})$, which leverages the query's unique subgraph context $\mathcal{G}_h$. This step assigns a dynamic, context-aware score to each

---

**Input**: query $\mathbf{q}(h, r, ?)$, model $\phi(\cdot)$, hyperparameters

**Output**: a vector of scores that measure the plausibility of different entities answering $\mathbf{q}$

1. Candidate rule generation from global rule base

2. For each rule $r \leftarrow \mathbf{r}_b$, predict $\phi(h, r, \mathbf{r}_b)$ from model $\phi(\cdot)$

3. Compute adjusted scores $\phi'$ using penalty $\lambda$, then update to confidence $w_i$ using softmax with temperature $T$.

4. For all the entities, calculate their potential to answer $\mathbf{q}$

    (a) Calculate a rule grounding score $ground(h, \mathbf{r}_b)$

    (b) Apply a binarization function to the grounding score $B(ground(h, \mathbf{r}_b))$

    (c) Compute $\mathbf{v}_{\text{ans}}$ using equation 2, which represents the entities potential to answer $\mathbf{q}$

5. Return $\mathbf{v}_{\text{ans}}$

---

Figure 2: Algorithm to answer a query $\mathbf{q}(h, r, ?)$

rule, moving beyond static metrics to assess its true relevance for the specific query. To mitigate the influence of overly broad rules which often introduce ambiguity in dense graphs, we calculate an adjusted score $\phi'(h, r, \mathbf{r}_b) = \phi(h, r, \mathbf{r}_b) - \lambda \log(n_{tails})$, where $n_{tails}$ represents the number of distinct tail entities reachable by the rule and $\lambda$ is a penalty coefficient (refer to Sec. 5.2 for analysis). Next, the context-aware scores are converted into a confidence distribution using a temperature-controlled softmax (Step 3). The weight $w_i$ for each rule is calculated as $w_i = \frac{\exp(\phi'_i/T)}{\sum_j \exp(\phi'_j/T)}$. The temperature $T$ is an important hyperparameter. As $T \to 0$, the softmax approximates an argmax function, concentrating all weight onto the single highest-scoring rule. This allows the model to rely solely on the *strongest contextual signal*.

Utilizing the rules and their context-aware score, Step 4 calculates the potential of each entity to answer the given query. If the score of an entity $e$ is high, it means that it is very probably that $(h, r, e)$ is a valid fact. The potential is measured using a score that aggregates the grounding score and the rules' local importance score. Each rule is grounded to produce an answer vector (Steps 4a). A naive grounding counts the number of paths from $h$ to other entities that can be reached by applying the rule $r \leftarrow \mathbf{r}_{b_i}$. Let $\mathbf{g}_i = \text{ground}(h, \mathbf{r}_{b_i})$. The value $\mathbf{g}_i[j]$ is the number of paths from $h$ to an entity $e_j$ when applying the rule $r \leftarrow \mathbf{r}_{b_i}$. If no such path exists, $\mathbf{g}_i[j] = 0$. This grounding operation can be efficiently conducted through sparse matrix multiplication.

A naive grounding score can be a noisy signal dominated by high-degree nodes. To mitigate this effect, we squash the path counts using the `tanh` function, controlled by a tanh scale hyperparameter $\tau$, $B(\mathbf{g}_i) = \tanh(\mathbf{g}_i/\tau)$. This provides a soft, saturated count that is controlled by the tanh scale hyperparameter $\tau$, preventing an unbounded influence from numerous paths.

Finally (Step 4c), a vector $\mathbf{v}_{\text{ans}}$ that captures all the entities' plausibility to answer the given query is computed as the weighted sum of all the binarized, grounded rule vectors as in equation 2.

$$\mathbf{v}_{\text{ans}} = \sum_i w_i \cdot B(\mathbf{g}_i) \quad (2)$$

From the values in $\mathbf{v}_{\text{ans}}$, we can rank all the entities. The rank of the true tail entity $t$ is then used to compute the MRR and Hits@k.

## 5 EXPERIMENTS: SLOGIC'S PERFORMANCE IN KGC

**Datasets**. We conduct our experiments on widely used benchmark datasets WN18RR (Dettmers et al., 2017), FB15k-237 (Toutanova & Chen, 2015), YAGO3-10 (Suchanek et al., 2007). For all datasets, we augment the data by adding inverse triplets $(t, r^{-1}, h)$ for each original triplet $(h, r, t)$, a common practice in the literature. The details about the datasets are described in Appendix A.

**Baselines**. SLogic is compared with 5 non-rule based approaches including TransE (Bordes et al., 2013), DistMult (Yang et al., 2014), ConvE (Dettmers et al., 2018), ComplEx (Trouillon et al., 2016), and RotatE (Sun et al., 2019), and 6 rule based approaches including AMIE (Galárraga

et al., 2013), Neural-LP (Yang et al., 2017), DRUM (Sadeghian et al., 2019), RNNLogic (Qu et al., 2021), RLogic(Cheng et al., 2022), and NCRL Cheng et al. (2023).

**Evaluation metrics.** We evaluate our model using Mean Reciprocal Rank (MRR), and Hits@k (k=1, 10) under the standard filtered setting (Bordes et al., 2013). For each test triple $(h, r, t)$, we evaluate by predicting $t$ for the forward query $(h, r, ?)$ and $h$ for the inverse query $(t, r^{-1}, ?)$. Rule-based systems frequently produce tied scores for many entities, which can lead to misleadingly optimistic ranks; for instance, a naive approach would assign a rank of 1 if no rules apply and all entities receive a score of zero. To robustly handle all such ties, we adopt the expected rank strategy from Qu et al. (2021) for tie-breaking, where the rank is calculated as $m + (n + 1)/2$; here, $m$ is the number of entities with a strictly higher score than the correct answer, and $n$ is the number of other entities sharing the same score. For queries with head entities not seen during training, we fall back to ranking answers based on the tail entity frequency for the given relation.

Further details on the experimental setting, including default setting of hyperparameters, hardware configuration, implementation specifics, and training procedures, are provided in Appendix B.

### 5.1 COMPARISONS WITH BASELINE METHODS

Table 1 shows the effectiveness of SLogic when comparing with the baselines. It shows that SLogic outperforms other methods on WN18RR and YAGO3-10. On FB15k-237, SLogic achieves competitive results (0.30 MRR), ranking second among rule-based approaches. The performance gap compared to sparse graphs like WN18RR is attributed to the high density of FB15k-237, where valid rules often possess high coverage (reaching many tail entities), introducing ambiguity that dilutes the discriminative power of context-aware scoring. A detailed analysis is in Appendix C.

The numbers for other systems are taken from Cheng et al. (2022; 2023). Since NCRL[†] employs a different evaluation protocol, where the rank is computed as $m + 1$, compared to the strategy used for all other methods. To ensure a fair comparison, we reran NCRL using our evaluation metric and reported these results as NCRL[*].

| | Models | WN18RR | | | FB15K-237 | | | YAGO3-10 | | |
|---|---|---|---|---|---|---|---|---|---|---|
| | | MRR | H@1 | H@10 | MRR | H@1 | H@10 | MRR | H@1 | H@10 |
| Non-rule based | TransE | 0.23 | 2.2 | 52.4 | 0.29 | 18.9 | 46.5 | 0.36 | 25.1 | 58.0 |
| | DistMult | 0.42 | 38.2 | 50.7 | 0.22 | 13.6 | 38.8 | 0.34 | 24.3 | 53.3 |
| | ConvE | 0.43 | 40.1 | 52.5 | **0.32** | 21.6 | 50.1 | 0.44 | 35.5 | 61.6 |
| | ComplEx | 0.44 | 41.0 | 51.2 | 0.24 | 15.8 | 42.8 | 0.34 | 24.8 | 54.9 |
| | RotatE | 0.47 | 42.9 | 55.7 | **0.32** | **22.8** | **52.1** | 0.49 | 40.2 | **67.0** |
| Rule-based Learning | AMIE | 0.36 | 39.1 | 48.5 | 0.23 | 14.8 | 41.9 | 0.25 | 20.6 | 34.3 |
| | Neural-LP | 0.38 | 36.8 | 40.8 | 0.24 | 17.3 | 36.2 | - | - | - |
| | DRUM | 0.38 | 36.9 | 41.0 | 0.23 | 17.4 | 36.4 | - | - | - |
| | RNNLogic* | 0.46 | 41.4 | 53.1 | 0.29 | 20.8 | 44.5 | 0.34 | 24.2 | 52.5 |
| | RLogic | 0.47 | 44.3 | 53.7 | 0.31 | 20.3 | 50.1 | 0.36 | 25.2 | 50.4 |
| | NCRL[†] | 0.67 | 56.3 | 85.0 | 0.30 | 20.9 | 47.3 | 0.38 | 27.4 | 53.6 |
| | NCRL* | 0.27 | 22.6 | 33.9 | 0.17 | 9 | 32.9 | 0.14 | 4 | 33.5 |
| | SLogic | **0.49** | **44.7** | **55.8** | 0.30 | 21.9 | 46 | **0.50** | **42.8** | 63.4 |

Table 1: SLogic vs. Baselines in KG completion task (**Bold**/Underlined numbers: best among all methods/best among all rule learning methods; '–': could not be run on our machine). SLogic results are reported as the mean over 5 random seeds with low standard deviation (WN18RR: $0.49\pm0.0007$, FB15k-237: $0.3 \pm 0.0006$, YAGO3-10: $0.50 \pm 0.0007$).

### 5.2 SENSITIVITY ANALYSIS AND ABLATION STUDY

This section analyzes the sensitivity of our model to several key hyperparameters and presents an ablation study by removing different components during the inference stage.

**Effect of positive and negative sampling ratio.** This analysis evaluates our model's sensitivity to the two parameters, $k_{pos}$ and $k_{neg}$. As shown in Table 2, the results demonstrate that the model is highly robust to different parameter settings. Performance remains stable within a narrow margin

across all tested configurations on all the datasets, with the configuration $k_{pos} = 5$ and $k_{neg} = 20$ consistently ranking among the best. The key practical implication of this robustness is that an exhaustive and computationally expensive search for optimal values is unnecessary. This is particularly valuable for larger datasets such as FB15K-237 and YAGO3-10, where memory constraints make more resource-intensive configurations infeasible. We repeated these sensitivity experiments with 3 random seeds and observed minimal variance ($\sigma \leq 0.002$), confirming that the sensitivity patterns shown in Table 2 are robust to initialization noise.

| $k_{pos}$ | WN18RR | | | FB15k-237 | | | YAGO3-10 | | |
|---|---|---|---|---|---|---|---|---|---|
| | $k_{neg} = 10$ | $k_{neg} = 20$ | $k_{neg} = 40$ | $k_{neg} = 10$ | $k_{neg} = 20$ | $k_{neg} = 40$ | $k_{neg} = 10$ | $k_{neg} = 20$ | $k_{neg} = 40$ |
| 1 | 0.4849 | 0.4827 | 0.4845 | 0.2937 | blue0.2941 | 0.2945 | 0.4865 | 0.4835 | 0.4975 |
| 5 | 0.4852 | 0.4871 | 0.4841 | 0.2974 | 0.2994 | - | 0.4986 | 0.5031 | 0.5011 |
| 10 | 0.4856 | 0.4880 | 0.4850 | 0.2979 | 0.2988 | - | 0.5036 | 0.5031 | - |

Table 2: Sensitivity test on the effect of sampling ratio ($k_{pos}$ vs. $k_{neg}$) on MRR. Darker cells indicate higher MRR. Certain configurations for FB15k-237 and YAGO3-10 (marked with -) were omitted due to excessive memory requirements.

**Effect of # of rules per query and subgraph hops.** Figure 3 shows that optimal hyperparameters depend strongly on KG structure. As in (a), the ideal number of inference rules varies widely: WN18RR peaks at 50–70 rules, YAGO3-10 at only 10, and FB15k-237 exhibits more volatile behavior. This suggests that for large graphs like YAGO3-10, the top-ranked rules are highly reliable and adding more only introduces noise. Figure 3(b) shows that a larger local context (more hops) brings more benefit on WN18RR although the performance increase is not obvious when having more than 2 hops. We cannot access a larger local context using more than 1-hop for the much denser graphs FB15k-237 and YAGO3-10.

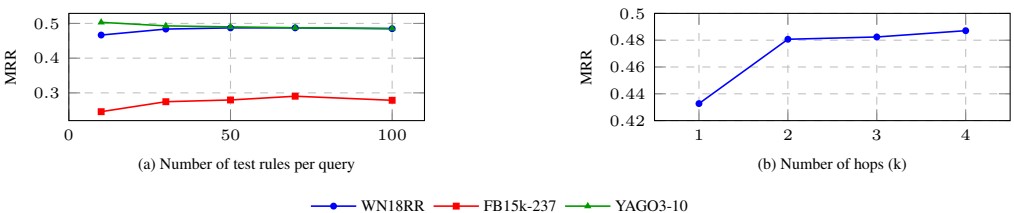

(a) Number of test rules per query    (b) Number of hops (k)

Figure 3: Sensitivity analysis. (Higher number of hops could not be run for FB15k-237 and YAGO3-10 on our machine.)

### 5.3 ABLATION STUDY AND EFFECT OF INFERENCE HYPERPARAMETERS

**Impact of query-dependent scoring.** To quantify the benefit of our dynamic scoring mechanism over traditional static approaches, we compared SLogic against a baseline variant, *SLogic-Static*. This baseline utilizes the identical rule mining and inference pipeline but relies solely on the global Wilson score for ranking, ignoring the query-specific subgraph context.

As shown in Figure 4a, SLogic consistently outperforms the static baseline, confirming the value of context-aware re-ranking. Notably, the performance gains are substantially larger for WN18RR (+24.4%) and YAGO3-10 (+25.4%) compared to FB15k-237 (+17.6%). Our analysis of the rule distributions reveals the cause: on FB15k-237, over 50% of the candidate rules already possess extremely high global confidence (Wilson score > 0.9), leaving little room for improvement via contextual re-scoring. In contrast, WN18RR and YAGO3-10 contain a higher proportion of rules with intermediate confidence (approx. 0.5), scenarios where dynamic, context-specific selection provides the most critical discriminative power.

**Ablation study of model components.** To validate the necessity of each architectural module, we conducted an ablation study on the YAGO3-10 dataset. We measured the performance impact (MRR drop) when removing four major components one by one.

Figure 4b summarizes the results. The significant drop observed when removing Hard Negative Sampling (-16.5%) and the Rule Encoder (-14.9%) indicates that these components form the foundational basis of the model's reasoning capability. Furthermore, the exclusion of the Subgraph En-

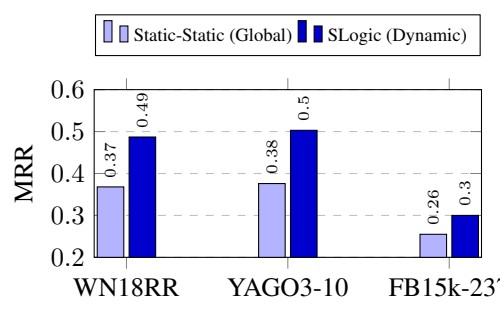

(a) Performance comparison: Dynamic vs. Static.

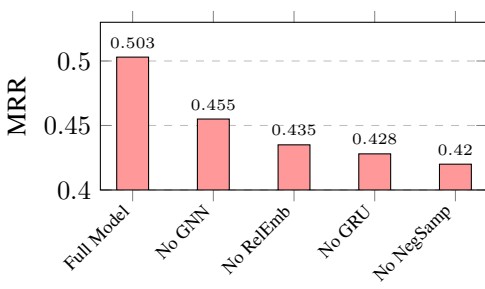

(b) Ablation study on YAGO3-10.

Figure 4: Impact of Dynamic Scoring (a) and Component Importance (b). SLogic achieves +17.6% to +25.4% gain over static baselines, and all neural components are essential.

coder (GNN) results in a 9.5% performance decrease, confirming that the GNN provides the critical "contextual lift" necessary to push performance beyond static baselines.

**Ablation study of inference components.** Figure 5 shows the effect of including/excluding the two components of (1) normalizing confidence score and (2) binarization of ground score. Including the normalization score calculation consistently improves the performance (from ✗ to low softmax temperature ($T = 0.5$)). When this component is included, the performance is not sensitive to the value of $T$. Regarding the binarization component, including it (✗) helps with YAGO3-10, but not the other two datasets.

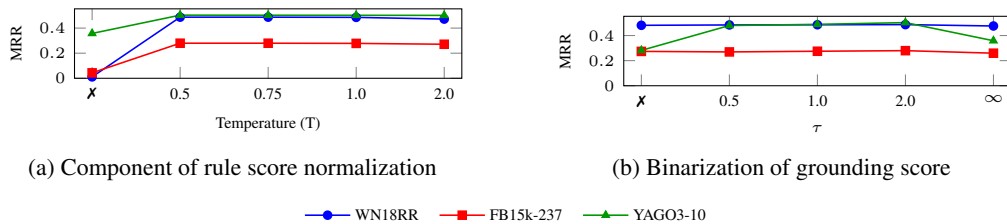

(a) Component of rule score normalization          (b) Binarization of grounding score

Figure 5: Ablation study of inference components and effect of inference hyperparameters. The ✗ symbol denotes that the component is disabled.

**Effect of rule coverage penalty** ($\lambda$). On the dense FB15k-237 dataset, introducing the coverage penalty with $\lambda \in [0.5, 0.7]$ increased MRR from 0.28 to 0.30 (+7%). This confirms that down-weighting high-coverage rules is critical for dense graphs. (See Appendix C for detailed diagnostic).

### 5.4 CASE STUDY: THE IMPACT OF LOCAL CONTEXT ON RULE SCORING

To demonstrate our model's ability to perform context-aware reasoning, we present a case study to compare the scoring of two rules on the YAGO3-10 dataset. We identified two distinct queries for the relation `isLocatedIn`, which is shortened to be `isL`. The rules and their global Wilson scores are held constant.

**Rule A:**`isL(X,Y):-isL(X,Z1),participatedIn(Z1,Z2),isL(Z2,Y)`(W-Score: 0.8545)
**Rule B:**`isL(X,Y):-isL(X,Z1),hasCapital(Z1,Z2),isL(Z2,Y)` (W-Score: 0.9578)

For the two queries (see Table 3), the different contexts come from the different head entities and their corresponding 1-hop subgraph.

As shown in Table 3, our model's final preference between these two rules flips depending on the local subgraph encoder in SLogic. Query 1 comes from the film industry, this context aligns better with the event-based rule (Rule A). Thus, our model after learning from the query-specific subgraphs gives it a higher local score (-0.3528) than Rule B despite Rule B's global score is higher. Query 2 on the other hand is more related to geographical and institutional information. Thus, Rule B receives a higher local score (0.0251) than Rule A.

|  | Case 1 | Case 2 |
|---|---|---|
| **Query** | `(Old_Shatterhand_(film),isL,?)` | `(U. of Alaska System,isL,?)` |
| **Subgraph Context** | Dominated by **film-industry** entities, e.g., actors (`Lex Barker`) and directors (`Hugo Fregonese`). | Dominated by **geographical** entities, e.g., cities (`Fairbanks, Alaska`) and countries (`United States`). |
| **Model's Score for A** | **-0.3528** | -1.6444 |
| **Model's Score for B** | -2.4848 | **0.0251** |

Table 3: A case study showing SLogic's contextual scoring capability.

## 5.5 EFFICIENCY ANALYSIS

For the rule-based method, we report the time required for rule collection across the different methods as well as the training time to obtain the performance shown in Table 1. Inference time is excluded from the comparison, since all methods employ comparable inference procedures.

| Method | Sub-component time | WN18RR | FB15k-237 | YAGO3-10 |
|---|---|---|---|---|
| SLogic | Mining time | 0.25 | 54 | 56 |
| | Negative sampling time | 149 | 189 | 826 |
| | Training time ($k_{pos} = 5, k_{neg} = 20$) | 552 | 892 | 291 |
| | **Total time ($k_{pos} = 5, k_{neg} = 20$)** | **701** | **1135** | **1173** |
| | Training time ($k_{pos} = 1, k_{neg} = 10$) | 50 | 75 | 25 |
| | **Total time ($k_{pos} = 1, k_{neg} = 10$)** | **199** | **318** | **907** |
| RNNLogic | Mining time | 88 | 279 | 87 |
| | Training time | 332 | 252 | 491 |
| | **Total time** | **420** | **513** | **578** |
| DRUM | **End to end training** | **55** | **717** | - |
| NCRL | **End to end training** | **6** | **126** | **336** |

Table 4: Running time (in minutes) across different methods and datasets ('–': Out of memory)

Table 4 shows that SLogic requires more training time than the baseline methods for all the datasets under the default setting. This overhead arises primarily because generating positive–negative sample pairs substantially increases the number of training instances. Under the default setting of $k_{\text{pos}} = 5$ and $k_{\text{neg}} = 20$, the resulting number of instances is approximately 100 times larger than the number of triplets in the original graph. In contrast, identifying negative samples is more time-consuming, as this step requires access to the entire graph. However, adopting an efficient setting ($k_{pos} = 1, k_{neg} = 10$) significantly reduces the training time, achieving a $\sim 3.5\times$ speedup on WN18RR and FB15k-237 with negligible MRR drop ($< 1.3\%$) and a $1.3\times$ speedup on YAGO3-10, offering a practical trade-off for resource-limited scenarios.

## 6 CONCLUSION AND DISCUSSIONS

We proposed a novel rule–based learning framework for KGC. Unlike existing approaches that rely on globally fixed rule confidences, our method leverages query contexts (subgraphs) to dynamically recalculate rule importance. This context-aware mechanism resolves uncertainty missed by static scores, enabling more accurate reasoning. SLogic outperforms other rule-based baselines and remains competitive with embedding methods, demonstrating its effectiveness by resolving structural ambiguity and accounting for rule coverage in dense graphs. However, SLogic has limitations. Training is more computationally expensive due to subgraph extraction and hard negative sampling, though sampling reduction offers a workable speed–accuracy trade-off. Finally, SLogic is limited to chain-like Horn clauses; supporting complex non-chain structures and exploring improved graph sampling methods are left for future work.

## REPRODUCIBILITY STATEMENT

To facilitate reproducibility, we provide an code repository at: `https://anonymous.4open.science/r/slogic-81FE/`. The repository contains the datasets used in our experiments, as well as scripts for data preprocessing, model training, and evaluation.

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

# A  DATASETS

This work utilizes three widely-used benchmark datasets for knowledge graph completion: WN18RR, FB15k-237 and YAGO3-10. These datasets are standard in the field as they have been curated to prevent test triple leakage from the training set. To support rule learning methodologies, we preprocess each knowledge graph by adding inverse triplets. The statistics of datasets are summarised in Table 5.

- WN18RR, introduced in Dettmers et al. (2017), a subset of the WN18 dataset, which is designed to be an intuitive dictionary and thesaurus for natural language processing tasks. In WN18RR, entities represent word senses and relations define the lexical connections between them.

- FB15k-237, introduced in Toutanova & Chen (2015), a frequently used benchmark dataset derived from Freebase. It is a large, online collection of structured data from various sources, including user-contributed wiki data.

- YAGO3-10, introduced in Suchanek et al. (2007), a subset of the large-scale semantic knowledge base YAGO3. It is constructed by integrating information from multiple authoritative sources, including Wikipedia, WordNet, and GeoNames.

| Dataset | #Entities | #Relations | #Train | #Validation | #Test |
|---------|-----------|------------|--------|-------------|-------|
| WN18RR | 40,943 | 11 | 86,835 | 3,034 | 3,134 |
| FB15k-237 | 14,541 | 237 | 272,115 | 17,535 | 20,466 |
| YAGO3-10 | 123,182 | 37 | 1,079,040 | 5,000 | 5,000 |

Table 5: Dataset statistics

# B  EXPERIMENTAL SETUP

## B.1  DEFAULT SETTING OF HYPERPARAMETERS

. For rule base construction, the length of rule body (or the depth of the DFS) $L$ is set to 5 (for WN18RR) and 3 (for both FB15K-237 and YAGO3-10). In subgraph extraction, the number of subgraph hops $k$ is set to be 4 for WN18RR dataset and 1 for both FB15K-237 and YAGO3-10 datasets. The hyperparameter $\alpha$ is 100 for all the datasets. For model training, the margin $\epsilon$ is set to be 1, $k_{pos}$ and $k_{neg}$ are set to be 5 and 20 respectively for all the three datasets. During inference stage, we set the default parameters as $N = 50$ for WN18RR and FB15k-237, and $N = 10$ for YAGO3-10. For all datasets, we use $T = 0.5$ and $\tau = 2.0$. Regarding the rule coverage penalty, we set $\lambda = 0.65$ for FB15k-237 and $\lambda = 0$ for WN18RR and YAGO3-10.

## B.2  HARDWARE

All experiments were conducted on a Dell PowerEdge R7525 server. This machine is equipped with 512 GiB of system RAM and powered by two AMD EPYC 7313 CPUs, providing a total of 32 physical cores (64 threads) running at a base clock speed of 3.0 GHz. For model training and inference, we utilized a single NVIDIA A100 GPU with 80 GiB of VRAM.

## B.3  IMPLEMENTATION DETAILS

Our model and training pipeline are implemented using PyTorch and PyTorch Geometric PyTorch Geometric (PyG) library (version 2.5.2) (Fey & Lenssen, 2019). Below, we provide detailed descriptions of the model architecture, data handling procedures, and training configuration.

**Model Architecture.** The SLogic model is a hybrid neural network composed of three main parts:

1. A **relation embedding layer** (`torch.nn.Embedding`) that provides dense vector representations for all relations in the knowledge graph. A designated padding index is used to handle variable-length rule bodies.

2. The **subgraph encoder** is a stack of Relational Graph Convolutional Network (`RGCNConv`) layers. We use 1 GNN layer for both FB15k-237 and YAGO3-10 dataset and 2 GNN layers for WN18RR, each followed by a ReLU activation and a dropout layer (p=0.5). The encoder takes the node feature matrix and the subgraph's edge information as input and produces final node embeddings. We extract two outputs: the embedding of the head node itself and a graph-level embedding computed via global mean pooling.

3. The **rule encoder** is a single-layer Gated Recurrent Unit (`torch.nn.GRU`) that processes the sequence of relation embeddings corresponding to a rule body. The final hidden state of the GRU is used as the rule's semantic embedding.

These components are integrated by a final scoring MLP. The feature vector for the MLP is a concatenation of: (1) the head node embedding from the GNN, (2) the graph-level embedding from the GNN, (3) the query relation embedding, (4) the rule body embedding from the GRU, and (5) a 4-dimensional vector of the rule's static statistics (support, confidence, Laplace confidence, and Wilson score). This combined vector is passed through a two-layer MLP with a ReLU activation and dropout to produce the final scalar score.

**Data handling and leakage prevention.** We use a custom PyTorch Geometric `Dataset` class to load the pre-computed subgraphs and training metadata. A critical aspect of our data loading process is the prevention of data leakage. During training, for a given triple $(h, r, t)$, the GNN must not have access to the direct edge $(h, r, t)$ in the subgraph $\mathcal{G}_h$, as this would allow it to solve the task trivially.

To prevent this, our `Dataset` class, during the loading of each individual sample, dynamically removes both the target edge $(h, r, t)$ and its corresponding inverse edge $(t, r^{-1}, h)$ from the subgraph's `edge_index` and `edge_attr` tensors before the subgraph is passed to the model. This ensures that the GNN must rely on the broader structural context rather than a simple edge-detection shortcut.

**Training details.** The model is trained end-to-end by minimizing a margin-based ranking loss (`torch.nn.MarginRankingLoss`) with a margin of $\epsilon = 1.0$. We use the Adam optimizer Kingma & Ba (2014) with a learning rate of $0.001$. For each training step, a batch of positive and negative rule pairs is processed. The rule bodies, which are sequences of relation IDs of variable length, are left-padded to the maximum length in the batch using our designated padding index. The model is trained for 5 epochs for all the datasets. The embedding dimensionality for both the relations and the RGCN layers was set to 128.

## C    IMPACT OF RULE COVERAGE (DIAGNOSTIC ANALYSIS)

In Section 5.2, we noted that SLogic performs differently on dense graphs (FB15k-237) compared to sparse ones. To understand this, we analyzed the relationship between the rule scores assigned by SLogic and the *rule coverage* ($n_{tails}$), defined as the number of distinct tail entities reachable by the rule.

Table 6 shows the average scores assigned to rules grouped by their coverage on FB15k-237. The model assigns significantly higher scores to high-coverage rules.

| Rule Coverage ($n_{tails}$) | Avg SLogic Score |
|---|---|
| Low (1-4 tails) | -1.12 |
| Medium-Low (4-19 tails) | -1.02 |
| Medium (19-73 tails) | -0.28 |
| Medium-High (73-339 tails) | 0.11 |
| High (339-4123 tails) | 0.57 |

Table 6: Relationship between rule coverage and model scores on FB15k-237.

While high-coverage rules are often valid, they are ambiguous for prediction because they generate hundreds of candidate entities. The high scores assigned to these rules overshadow more specific, discriminative rules. By introducing the penalty term $-\lambda \log(n_{tails})$, we explicitly dampen the influence of these high coverage rules. Our experiments confirmed that this penalty yields a 7%

performance improvement on FB15k-237, whereas for sparse graphs like WN18RR and YAGO3-10, where rules are naturally specific, $\lambda = 0$ remains optimal.

## D USE OF LARGE LANGUAGE MODELS (LLMs)

ChatGPT and Gemini were used to correct grammatical errors and enhance the clarity of the writing.

