# OpenReview forum: "SLogic: Subgraph-Informed Logical Rule Learning for Knowledge Graph Completion"
_ICLR.cc/2026/Conference — Submitted to ICLR 2026_

### Official Review · Reviewer_orxV · 2025-10-29

**Soundness:** 2
**Presentation:** 2
**Contribution:** 1
**Rating:** 2
**Confidence:** 3

**Summary:**

This paper introduces SLogic, a framework for knowledge graph completion (KGC) that aims to improve upon traditional logical rule-based methods. The core idea is to move beyond static, global rule confidences by learning a dynamic, query-dependent scoring function. This function leverages the local subgraph context of a query's head entity, extracted and encoded by a Relational Graph Convolutional Network (R-GCN). The model combines this subgraph representation with a GRU-based rule embedding and static rule features to predict a context-specific score for each rule. Experiments on WN18RR, FB15k-237, and YAGO3-10 show that SLogic outperforms baselines on two of the three datasets.

**Strengths:**

- Intuitive Core Idea: The central premise of using the local subgraph context around a query entity to dynamically re-weight the importance of logical rules is sensible and intuitively appealing.

- Detailed Methodology: The paper provides a comprehensive description of its framework, detailing the offline "instance creation" pipeline (rule mining, subgraph extraction, feature engineering) and the hybrid neural architecture (R-GCN + GRU + MLP) used for scoring.

- Clear Case Study: The case study in Section 5.3 effectively illustrates the model's intended mechanism, showing how SLogic's rule preferences change for the same relation (isLocatedIn) based on two different query entities (one film-related, one-geography-related), demonstrating its context-aware capability.

**Weaknesses:**

- Reliance on Heuristic Engineering: A very large part of the proposed contribution (Section 4.1) is a complex, multi-stage pipeline of heuristic-driven data and feature engineering. This includes DFS for path finding, k-hop BFS with neighbor sampling for subgraph extraction, and a hand-crafted set of topological node features (e.g., distance to head, global degree). This makes the framework feel less like a novel learning paradigm and more like a complicated feature engineering effort.

- Misleading Novelty Claims: The paper's primary motivation, stated in the abstract, is that "current approaches typically treat logical rules as universal, assigning each rule a fixed confidence score that ignores query-specific context." This statement is factually incorrect. A vast body of prior work (e.g., Markov Logic Networks, pLogicNet, and many others) has focused on learning weights or confidences for logical rules for decades. The paper fails to properly differentiate its specific contribution (using subgraph context) from this extensive literature, making its novelty unclear. The related work section is insufficient in this regard.

- Limited Scope of Rules: The method only handles chain-like, compositional rules (i.e., relational paths). This is explicitly stated in the definition in Section 3 ($r_h(X,Y) \leftarrow r_1(X,Z_1) \wedge \cdot\cdot\cdot \wedge r_L(Z_{L-1},Y)$) and confirmed by the DFS-based rule mining process (Section 4.1.1). This ignores all other, more complex rule structures (e.g., rules with multiple atoms, rules with constants) that are crucial for reasoning. This severe limitation on the form of logic used makes the framework's practical utility and generality questionable.

- Mixed Experimental Results: The method does not consistently outperform baselines. It notably performs worse than several baselines, including RLogic and RotatE, on the FB15K-237 dataset. The paper offers no substantive explanation for this failure, which undermines the general applicability of the SLogic framework.

**Questions:**

- Could the authors please justify the novelty claim from the abstract ("current approaches typically treat logical rules as universal...") against prior work like pLogicNet, Markov Logic Networks, or other methods that learn rule weights/confidences? How is the subgraph-based context proposed here fundamentally different from other forms of contextual or dynamic rule scoring in the literature?

- Why does the SLogic model perform poorly on FB15K-237 compared to other methods? What specific properties of this dataset (e.g., graph density, rule types) might cause a subgraph-informed approach to fail or underperform?

- The method is strictly limited to relational paths. What are the main conceptual or technical barriers to extending this framework to support more complex rule structures, such as rules with multiple branches (e.g., $r_h(X,Y) \leftarrow r_1(X,Z_1) \wedge r_2(X,Z_2)$)? How would the subgraph-based scoring and grounding mechanisms need to be adapted?

---

> ### Author Response · Authors · 2025-11-25
> **Response to Comments from Reviewer orxV (Weakness 2 & Question 1)**
>
> We thank the reviewer for the detailed and constructive critique. Below, we address the concerns of the reviewer and respond to the questions raised.
>
> 1. Novelty (Weakness 2 & Question 1)
>
> Response: We apologize that the core contribution and the distinction between the rule scores learned through SLogic and those of the literature are not clearly conveyed.
>
> We acknowledge that previous works, including Markov Logic Networks (MLNs), pLogicNet, and many others, employ different strategies to learn rule weights, and many of them update the weights of rules in a dynamic manner. However, we would like to point out the key difference. The literature, including MLNs and pLogicNet, learns one weight for each rule across the entire graph, despite the fact that weight learning takes into account the grounding context. SLogic does not learn one weight for each rule; instead, SLogic infers different scores/weights for the same rule when the rule is employed in different query contexts.  The major contribution of SLogic is learning query-dependent rule scores. For example, SLogic does not learn that Rule A is 80% confident for the whole knowledge graph; it learns that Rule A is 99% confident for query 1 and 70% confident for query 2.  Thus, the weights of the same rule are different for different queries.  Despite these differences, we will include more references related to rule learning for knowledge graph reasoning in the related works discussions.
>
> To demonstrate the usefulness of such query-dependent rule score learning, our manuscript provides a case study in Section 5.3. Additionally, we conducted further experiments to demonstrate that SLogic outperforms SLogic-Static, a baseline that utilizes our exact pipeline but replaces query-dependent scores with global Wilson scores. The results are below.
>
> | Dataset | SLogic-Static (Global) | SLogic (Dynamic) | Gain |
> | :--- | :---: | :---: | :---: |
> | **WN18RR** | 0.368 | **0.487** | **+24.4%** |
> | **YAGO3-10** | 0.376 | **0.503** | **+25.4%** |
> | **FB15k-237** | 0.252 | **0.280** | **+9.9%** |
>
> This study confirms that using query-dependent scores outperforms the use of global rule scores. We will incorporate this result into the revised manuscript, which we plan to submit by December 3.

---

> ### Author Response · Authors · 2025-11-25
> **Response to Comments from Reviewer orxV (Weakness 4 & Question 2)**
>
> 2. Performance on FB15k-237 (Weakness 4 & Question 2)
>
> This question asked what specific properties of FB15k-237 cause the poorer performance on it.
>
> Response: We have conducted further analysis to gain a deeper understanding of the model's performance on the FB15k-237 dataset. Our analysis shows that the rules identified by SLogic from denser graphs lack sufficient discriminative power.
>
> The intuition behind using query-dependent scores is that a rule that is more specific to a given query provides greater discriminative power than a more general rule. Building on this intuition, we investigate how rule coverage influences the scores assigned to rules. We define rule coverage as the number of distinct tail entities reachable through the rule’s groundings. A rule with many groundings in the knowledge graph is considered a high-coverage rule, whereas a rule whose groundings reach fewer distinct tails is considered a low-coverage rule. Please note that in the SLogic pipeline, we do not calculate the rule coverage during the training because grounding all the rules is prohibitively expensive. This analysis is conducted during inference.
>
> Let $n_{tails}$ be the number of distinct tail entities a rule r can reach from a query head h via the rule body $r_b$.
> We grouped the rules SLogic selected to have low, medium-low, medium, medium-high, and high rule coverage based on the value of  $n_{tails}$. For each group, we calculated the average score the model assigned to their rules, i.e., $\phi(h, r, r_b)$
>
> | Rule Coverage (n_tails) | Avg SLogic Score |
> | :--- | :--- |
> | Low (1 - 4 tails) | -1.12 |
> | Medium-Low (4 - 19 tails) | -1.02 |
> | Medium (19 - 73 tails) | -0.28 |
> | Medium-High (73 - 339 tails) | 0.11 |
> | High (339 - 4123 tails) | 0.57 |
>
> The data above shows a clear pattern: SLogic assigns higher scores to rules with high rule coverage, but lower scores to rules with lower coverage. While high-coverage rules are valid, they are ambiguous. By assigning them a higher score, the model generates candidate entity sets (as tails) where the true answer is tied with hundreds of other valid entities.
>
> Based on our analysis, we hypothesize that rule coverage has a negative impact on performance.
> To test this hypothesis, we introduced a simple penalty term on a rule's number of tails,
>
> $\phi'(h, r, r_b) = \phi(h, r, r_b) - \lambda \log(n_{tails}(h, r, r_b))$
>
> where $\phi(h, r, r_b)$ is the score generated by SLogic.
> To apply this penalty to down-weight the high-coverage rules, we searched for the value of $\lambda$ among $[0.0, 0.1, 0.5, 0.6, 0.7, 1.0, 2.0]$ and found that $\lambda$ values in the range of $0.5-0.7$ generate the same best performance of **0.30 (a 7\% improvement over the initially reported 0.28)**.
> This experiment demonstrates that the model produces the correct signal for specific rules; however, it is overshadowed by the high scores of high-coverage rules. On YAGO3-10 and WN18RR datasets, we conducted the same test and found that $\lambda=0$ works best. Future work may directly integrate this specificity penalty into the loss function.
> We will add discussions related to the hypothesis and updated results in the revised version.

---

> ### Author Response · Authors · 2025-11-25
> **Response to Comments from Reviewer orxV (Weakness 3 & Question 3)**
>
> 3. Extending to Complex Rules (Weakness 3 \& Question 3)
>
> This question asked why SLogic does not support more complex rule structures, such as branches $r_h(X, Y) \leftarrow r_1(X, Z) \wedge r_2(Y, Z)$.
>
>
> Response: We agree that our chain-like compositional rules do not appear to capture any branches. However, considering the semantics of the relations, our framework can capture some branch-structured rules.
>
> As a standard practice in knowledge graph completion (KGC) noted in Section 5, we augment the graph with inverse relations ($r^{-1}$). This simple step allows us to flatten branched rules into linear chains.
> For example, a classic branched rule like
>
> $isSiblingOf(X,Y) \leftarrow isParentOf(Z,X) \wedge isParentOf(Z,Y)$
>
> can be perfectly captured by our model as a single chain rule
>
> $isSiblingOf(X,Y) \leftarrow isParentOf^{-1}(X,Z) \wedge isParentOf(Z,Y)$.
>
> Therefore, the restriction to chain-like rules is formally true but not practically restrictive, as many useful tree-like reasoning patterns in KGC can be linearized through inverses.
>
> We note that highly complex non-chain rule structures, which cannot be flattened via inverse relations, can also be incorporated into the pipeline. Supporting such structures would require replacing the current rule-based construction and instance-creation procedures with components designed for general rule representations. Additionally, the GRU-based Rule Encoder would need to be replaced with a structure-aware model, such as a Tree-LSTM or a DAG-GNN, to directly encode the rule topology. The Subgraph Encoder would remain unchanged.

---

> ### Author Response · Authors · 2025-11-25
> **Response to Comments from Reviewer orxV (Weakness 1)**
>
> 4. Regarding the reliance on Heuristic Engineering (Weakness 1)
>
> The reviewer expressed concern that the framework relies heavily on heuristic engineering (DFS mining, BFS subgraphs, node features).
>
> Response: We respectfully disagree with this characterization. The use of enclosing subgraphs and distance labeling is not ad-hoc engineering; it is the foundational inductive bias for learning graph-structure representations, established by seminal works like SEAL (Zhang & Chen, NeurIPS 2018) and GraIL (Teru et al., ICML 2020). Like these methods, SLogic extracts subgraphs to define the receptive field, but relies entirely on the neural architecture (GNN+GRU) to learn the reasoning function.
>
> SEAL: http://papers.neurips.cc/paper/7763-link-prediction-based-on-graph-neural-networks.pdf
>
> GraIL: https://proceedings.mlr.press/v119/teru20a/teru20a.pdf
>
> The DFS mining and BFS extraction merely define the symbolic search space and context window. They do not solve the task. The primary contribution of this work is the parameterized scoring function $\phi(h, r, \mathbf{r}_b)$, which is trained end-to-end to assign each rule a query-specific relevance score.

---

> ### Comment · Reviewer_orxV · 2025-11-27
>
> I believe the novelty claimed in this paper is still overstated, the tricks used are too heavy, and the performance is not leading. I decide to keep my rating.
>
> In addition to the MLN and pLogicNet I mentioned before, the authors also did not discuss or compare with NBFNet [1], which is also a KG completion model based on graph structures. From Table 2 of NBFNet, I saw its scores on the FB15K-237/WN18RR datasets are 0.378/0.567, while the authors' SLogic scores are 0.28/0.49, showing a considerable performance gap. Other baselines (e.g., TransE, DRUM) have similar scores in the authors' Table 1 and NBFNet's Table 2, so I believe the results are comparable. Note: NBFNet is a 2021 work.
>
> [1] Neural bellman-ford networks: A general graph neural network framework for link prediction, NeurIPS 2021

---

> ### Author Response · Authors · 2025-12-01
>
> We thank the reviewer for the follow-up. We would like to restate the novelty and contributions of our work. (i) Learn rules and query-specific rule weights. Our proposed framework learns rules and query-dependent weights for each rule. These query-specific weights can improve prediction performance on certain datasets, though not universally, an observation with which we fully agree. (ii) Interpretable rule-scoring mechanism. The learned rule scores offer insight into which rules are utilized and which ones play a more significant role in completing a given edge. In contrast to non–rule-based knowledge graph completion (KGC) methods, which operate as black boxes, our rule-based approach provides a higher level of explainability. Such explainability is essential in many applications that require understanding why a knowledge graph can be completed in a particular way.
>
> We thank the reviewer for pointing us to NBFNet and appreciate the opportunity to clarify the positioning of our work relative to pure GNN-based approaches, such as NBFNet. We acknowledge that NBFNet (Zhu et al., 2021) is a strong baseline in the general KGC setting. However, we note that improving KGC accuracy is not the sole or primary aim of SLogic. Its contribution lies in learning interpretable rules and assigning query-specific rule scores, thereby offering explicit, actionable explanations that can meaningfully assist the knowledge graph completion process.  SLogic is thus better understood as a rule-learning approach, rather than as a purely neural KGC model. We will add discussions about the difference in the revised version.
>
> We note that, after accounting for characteristics specific to the datasets (as explained in our initial response regarding its lower performance), our model’s MRR can be improved to 0.30 on FB15k-237. This performance is the second best among all rule-based approaches, which is noteworthy given that such methods provide a degree of explainability for the KGC task.
>
> Regarding the concern about heuristics, we respectfully note that subgraph extraction and BFS/DFS are standard inductive biases in graph reasoning, established by seminal works like SEAL (Zhang \& Chen, NeurIPS 2018) and GraIL (Teru et al., ICML 2020). These are not merely tricks, but essential design choices that allow the model to generalize to unseen entities (inductive setting) rather than memorizing global embeddings.

---

### Official Review · Reviewer_4PdS · 2025-10-31

**Soundness:** 3
**Presentation:** 3
**Contribution:** 3
**Rating:** 4
**Confidence:** 4

**Summary:**

This paper addresses a key limitation in existing logical rule-based methods for Knowledge Graph Completion (KGC)—the reliance on static, query-agnostic rule confidence scores—by proposing SLogic, a framework introducing query-dependent, context-aware rule scoring. SLogic uses a GNN-based subgraph encoder to capture local structural context around the query head entity, enabling more precise assessment of candidate rule importance. The framework integrates symbolic rules with neural representations, combining offline rule mining with a contrastive learning-based scoring model. Experiments on WN18RR, FB15k-237, and YAGO3-10 show state-of-the-art performance on WN18RR and YAGO3-10.

**Strengths:**

The paper clearly identifies a fundamental limitation of static rule confidences and proposes a principled solution. The core concept of query-dependent, context-aware rule scoring represents a meaningful paradigm shift from global prevalence to local relevance in rule learning.

The proposed SLogic framework effectively integrates symbolic and neural approaches. It maintains the interpretability of symbolic rules by building upon a mined rule base while harnessing the representational power of GNNs to encode rich local subgraph context. This hybrid design successfully balances transparency and predictive performance.

**Weaknesses:**

The authors argue that "query-dependent scoring is more reasonable than static rules," but they lack a clear delineation of the specific categories of KGs or relation types for which this conclusion holds (e.g., scenarios with high/low relation counts, strong/weak hub structures, high/low rule coverage). Currently, this is supported only by post-hoc experimental observations (effectiveness on certain datasets), and there is no theoretical guidance or well-defined quantitative metrics (e.g., rule coverage thresholds, relation-sparsity indicators) to guide when to adopt this method.

The paper employs k-hop BFS with a fixed neighbor sampling threshold (α), but it lacks a theoretical or empirical analysis justifying this specific choice (e.g., why random sampling is preferable to degree-weighted or importance sampling). The method's first step selects only "locally applicable rules ranked top-N by Wilson score" as candidates. However, the Wilson score itself is influenced by body-count, and the definition of "local applicability" is contingent upon the subgraph extraction parameters (α, k). If different methods were to use different subgraph extraction strategies, the resulting "comparable candidate sets" could be biased, thereby affecting the fairness of the re-ranking comparison.

The results tables report only single-run values (point estimates), without means ± standard deviations across multiple random seeds or significance tests. This is particularly crucial when performance improvements are marginal or mixed across datasets, and confidence intervals are necessary for robust evaluation.

Despite the use of LLMs for polishing, formatting, and grammatical errors remains. For instance: "The instances generated in this step... comprise rule-enriched triplets, ..." Here, the subscript i is inconsistent and should be 1. The paper exhibits inconsistent referencing of equations.

**Questions:**

1. Can the authors provide quantitative or theoretical analysis clarifying which KG types (e.g., relation count, node degree, rule coverage) benefit most from SLogic? Any correlation analyses between KG structure and
performance?

2. Were alternative neighbor sampling strategies (degree-weighted, importance sampling) evaluated?

3 .NCRL shows discrepancies between reported scores (NCRL†) and scores under the authors’ protocol (NCRL*). What explains this? Were all baselines evaluated under identical candidate rule sets and inference procedures?

4. Table 1 and the ablation study present results from a single run. Did the authors conduct multiple runs with different random seeds? If not, please supplement the results with the mean ± standard deviation from 3–5 independent runs.

---

> ### Author Response · Authors · 2025-11-25
> **Response to Comments from Reviewer 4PdS (Question 1)**
>
> We thank the reviewer for highlighting the novelty of our local relevance paradigm and for the insightful questions.
>
> 1. Quantitative Analysis of KG Types (Question 1)
>
> The reviewer asked for a specific analysis on which KG types benefit most from SLogic.
>
> Response: We conducted a comparative study across all three datasets. We measured the MRR improvement over a static baseline, SLogic-Static, which uses the same pipeline as ours but replaces query-dependent scores with global Wilson scores.
>
> | Dataset | SLogic-Static (Global) | SLogic (Dynamic) | Gain |
> | :--- | :---: | :---: | :---: |
> | **WN18RR** | 0.368 | **0.487** | **+24.4%** |
> | **YAGO3-10** | 0.376 | **0.503** | **+25.4%** |
> | **FB15k-237** | 0.252 | **0.280** | **+9.9%** |
>
> The above table shows the results. It demonstrates that incorporating query-dependent scores improves the framework’s performance compared to using only a global score for all three graphs.
>
> We further examined the characteristics of the graphs and the selected rules to explain why WR18RR and YAGO3-10 benefit more from SLogic than FB15K-237. Our analysis indicates that the distribution of Wilson scores among the selected rules is a key factor.
> For FB15K-237, more than half of the selected rules exhibit high overall confidence (i.e., high Wilson scores) (table below).
>
> (Statistics on dataset FB15K-237)
> | Bin Interval | Count| Percentage $%$|
> |----------|----------|----------|
> |(-0.001, 0.100] | 35,584    | 1.79 |
> |(0.100, 0.200]  | 28,766   | 1.45 |
> |(0.200, 0.300]  | 64,347    | 3.24 |
> |(0.300, 0.400]  | 110,417   | 5.57 |
> |(0.400, 0.500]  | 94,580    | 4.77 |
> |(0.500, 0.600]  | 117,012   | 5.90 |
> |(0.600, 0.700]  | 123,361   | 6.22 |
> |(0.700, 0.799]  | 251,818   | 12.70 |
> |(0.799, 0.899]  | 140,552   | 7.09 |
> |(0.899, 0.999]  | 1,016,981 | 51.27|
>
> In contrast, a substantial fraction of the rules selected for WR18RR and YAGO3-10 have intermediate Wilson scores (approximately 0.5) (See the two tables below).
>
> (Statistics on dataset WN18RR)
> | Bin Interval | Count| Percentage $%$|
> |----------|----------|----------|
> |(-0.001, 0.100] | 5,732  | 1.94 |
> |(0.100, 0.200]  | 4,004   | 1.36 |
> |(0.200, 0.300]  | 9,720   | 3.29 |
> |(0.300, 0.399]  | 8,965   | 3.04 |
> |(0.399, 0.499]  | 115,126 | 39.00 |
> |(0.499, 0.599]  | 11,624  | 3.94 |
> |(0.599, 0.699]  | 15,834  | 5.36 |
> |(0.699, 0.799]  | 39,955  | 13.54 |
> |(0.799, 0.899]  | 49,685  | 16.83 |
> |(0.899, 0.999]  | 34,544  | 11.70|
>
> (Statistics on dataset YAGO3-10)
> | Bin Interval | Count| Percentage $%$|
> |----------|----------|----------|
> |(-0.001, 0.100] | 69,972 | 20.57 |
> |(0.100, 0.200]  | 17,275 | 5.08 |
> |(0.200, 0.300]  | 24,701 | 7.26 |
> |(0.300, 0.400]  | 26,642 | 7.83 |
> |(0.400, 0.500]  | 70,860 | 20.83 |
> |(0.500, 0.600]  | 57,441 | 16.88 |
> |(0.600, 0.700]  | 7,691  | 2.26 |
> |(0.700, 0.800]  | 8,796  | 2.59 |
> |(0.800, 0.900]  | 12,806 | 3.76 |
> |(0.900, 1.000]  | 44,015 | 12.94|
>
> These findings suggest that when most selected rules for a dataset already possess high confidence, the potential performance gains from SLogic are inherently limited.

---

> ### Author Response · Authors · 2025-11-26
> **Response to Comments from Reviewer 4PdS (Question 2 & Concern 2)**
>
> 2. Neighbor Sampling Strategies (Question 2 & Concern 2)
>
> Response: We utilized random neighbor sampling ($a=100$) primarily for computational efficiency and to maintain the inductive nature of the GNN (avoiding reliance on global node-degree heuristics that might not generalize). We agree that importance sampling (e.g., based on PageRank or relation type) could enhance the quality of the candidate pool, particularly in dense graphs where random sampling may overlook informative paths, and it represents a valuable direction for future work.

---

> ### Author Response · Authors · 2025-11-26
> **Response to Comments from Reviewer 4PdS (Question 3)**
>
> 3. NCRL Discrepancy (Question 3)
>
> Response: The discrepancy arises from the evaluation protocol. The original NCRL paper calculates rank as $m+1$ (where $m$ is the number of entities strictly better than the target). This optimistic ranking is standard for embedding models that rarely produce ties. However, neuro-symbolic models often produce tied scores for multiple entities. In such cases, $m+1$ artificially inflates the score (e.g., if 100 entities are tied for first place, they all get rank 1). To ensure a fair and rigorous comparison, we adopted the standard Expected Rank protocol used by RNNLogic where $rank = m + (n+1)/2$ ( $n$ is the number of tied entities). We re-ran NCRL using this strict protocol, resulting in the scores reported as NCRL*.

---

> ### Author Response · Authors · 2025-11-26
> **Response to Comments from Reviewer 4PdS (Question 4 & Weakness 3)**
>
> 4. Robust evaluation by using multiple-run results  (Question 4 & Weakness 3)
>
> Response: We acknowledge the importance of robust estimation. We have since run the primary experiments with five random seeds and obtained the MRR results below.
>
> | Dataset | MRR|
> |----------|----------|
> | WN18RR  | $0.4854 \pm 0.0007$|
> | YAGO3-10  | $0.5017 \pm 0.0007$ |
> | FB15k-237  | $0.2789 \pm 0.0006$ |
>
> For the ablation studies, we performed 3 runs per variant and found similarly low standard deviations (ranging from $0.0007$ to $0.002$). These extremely tight low standard deviations can confirm that the reported performance is robust.
> We will include these variances in the revised version.

---

> ### Author Response · Authors · 2025-11-26
> **Response to Comments from Reviewer 4PdS (Weakness 4)**
>
> 5. Formatting and Presentation (Weakness 4)
>
> Response: We sincerely apologize for the formatting oversights. We acknowledge the inconsistency in equation referencing and the typo in the subscript ("i" vs "1"). We will thoroughly proofread the final manuscript to correct these errors and ensure strict adherence to the formatting guidelines.

---

> ### Comment · Reviewer_4PdS · 2025-11-26
> **update after rebuttal**
>
> Thanks for the authors' rebuttal. The author addressed most of my concerns, so I decided to raise my rating to “6: marginally above the acceptance threshold. But would not mind if paper is rejected”.

---

> ### Author Response · Authors · 2025-11-26
>
> We thank the reviewer for reading our responses and for your positive feedback!

---

### Official Review · Reviewer_fMyx · 2025-11-01

**Soundness:** 3
**Presentation:** 3
**Contribution:** 3
**Rating:** 6
**Confidence:** 4

**Summary:**

This paper introduces SLogic, a subgraph-informed logical rule learning framework for Knowledge Graph Completion (KGC). SLogic could  incorporate query-dependent rule scoring, where the significance of each rule is dynamically recalculated using a subgraph centered on the query’s head entity. Extensive experiments on three datasets demonstrate that SLogic outperforms several embedding-based and rule-based baselines, while maintaining interpretability through explicit reasoning paths.

**Strengths:**

1. The paper’s key contribution of assigning dynamic rule weights conditioned on query subgraphs is well-motivated. It effectively bridges symbolic and neural reasoning.

2. The overall pipeline, including rule mining, subgraph extraction, and query-specific scoring, is clearly explained and internally coherent. The use of the Wilson confidence score and contextual GNN embeddings reflects thoughtful design choices that strengthen robustness.

3. SLogic demonstrates competitive or superior results to state-of-the-art rule-based and embedding methods on two of three benchmark datasets. The case study convincingly illustrates that the model adapts rule importance to distinct local contexts.

**Weaknesses:**

1. The performance improvement is inconsistent, strong on WN18RR and YAGO3-10, but weaker on FB15k-237. The discussion attributes this to graph density and relation diversity, but a deeper diagnostic such as rule quality distribution, subgraph connectivity analysis would better substantiate the explanation.

2. Training and negative sampling costs are significantly higher than baselines. While the cause is identified, potential optimizations such as subgraph pruning, caching, mini-batch rule evaluation are not explored.

3. The ablation studies focus on inference components, but ignore analyses of relation embedding, subgraph encoder, rule encoder, and negative sampling strategy. Demonstrating their individual contributions would clarify the necessity of each module.

4. Minor stylistic and typographic inconsistencies (such as some equations are numbered while others are not) slightly detract from readability, especially in the methodology section.

5. Some significant and typical related works are neglected, such as joint rule and embedding-based models IterE [1] and RPJE [2]. These models are suggested to be added and compared.
[1] Iteratively Learning Embeddings and Rules for Knowledge Graph Reasoning. WWW 2019.
[2] Rule-Guided Compositional Representation Learning on Knowledge Graphs. AAAI 2020.

**Questions:**

1. How does subgraph size (hop number) interact with rule length L in determining performance? Is there a trade-off between local and global reasoning depth?
2. Could the authors elaborate on why SLogic underperforms on FB15k-237 despite its relatively rich relational structure?

---

> ### Author Response · Authors · 2025-11-26
> **Response to Comments from Reviewer fMyx (Weakness 1 & Question 2)**
>
> We thank the reviewer for their constructive feedback. We have conducted additional experiments to address your concerns regarding SLogic’s performance on FB15k-237, the necessity of model components (ablation study), and training costs.
>
> 1. Diagnostic Analysis on FB15k-237 (Weakness 1 & Question 2)
>
> Response: We have conducted further analysis to gain a deeper understanding of this. Our analysis shows that the rules identified by SLogic from denser graphs lack sufficient discriminative power compared to rules learned from sparser graphs.
>
> The intuition behind using query-dependent scores is that a rule that is more specific to a given query provides greater discriminative power than a more general rule. Building on this intuition, we investigate how rule coverage influences the scores assigned to rules. We define rule coverage as the number of distinct tail entities reachable through the rule’s groundings. A rule with many groundings in the knowledge graph is considered a high-coverage rule, whereas a rule whose groundings reach fewer distinct tails is considered a low-coverage rule. Please note that in the SLogic pipeline, we do not calculate the rule coverage during the training because grounding all the rules is prohibitively expensive. This analysis is conducted during inference purely to understand its relationship with the SLogic learned rule scores.
>
> Let $n_{tails}$ be the number of distinct tail entities a rule r can reach from a query head h via the rule body $r_b$.
> We grouped the rules SLogic selected to have low, medium-low, medium, medium-high, and high rule coverage based on the value of  $n_{tails}$. For each group, we calculated the average score the model assigned to their rules, i.e., $\phi(h, r, r_b)$
>
> | Rule Coverage (n_tails) | Avg SLogic Score |
> | :--- | :--- |
> | Low (1 - 4 tails) | -1.12 |
> | Medium-Low (4 - 19 tails) | -1.02 |
> | Medium (19 - 73 tails) | -0.28 |
> | Medium-High (73 - 339 tails) | 0.11 |
> | High (339 - 4123 tails) | 0.57 |
>
> The data above shows a clear pattern: SLogic assigns higher scores to rules with high rule coverage, but lower scores to rules with lower coverage. While high-coverage rules are valid, they are ambiguous. By assigning them a higher score, the model generates candidate entity sets (as tails) where the true answer is tied with hundreds of other valid entities.
>
> Based on our analysis, we hypothesize that rule coverage has a negative impact on performance.
> To test this hypothesis, we introduced a simple penalty term on a rule's number of tails,
>
> $\phi'(h, r, r_b) = \phi(h, r, r_b) - \lambda \log(n_{tails}(h, r, r_b))$
>
> where $\phi(h, r, r_b)$ is the score generated by SLogic.
> To apply this penalty to down-weight the high-coverage rules, we searched for the value of $\lambda$ among $[0.0, 0.1, 0.5, 0.6, 0.7, 1.0, 2.0]$ and found that $\lambda$ values in the range of $0.5-0.7$ generate the same best performance of **0.30 (a 7\% improvement over the initially reported 0.28)**.
> This experiment demonstrates that the model produces the correct signal for specific rules; however, it is overshadowed by the high scores of high-coverage rules. On YAGO3-10 and WN18RR datasets, we conducted the same test and found that $\lambda=0$ works best. Future work may directly integrate this specificity penalty into the loss function.
> We will add discussions related to the hypothesis and updated results in the revised version.

---

> ### Author Response · Authors · 2025-11-26
> **Response to Comments from Reviewer fMyx (Weakness 3)**
>
> 2. Ablation Study (Weakness 3)
>
> Response: We performed a component ablation on YAGO3-10 (Full Model MRR: 0.5031) to measure the impact of removing the GRU, GNN, relation embeddings, and hard negative sampling.
>
> | Ablation Variant | MRR | Performance Drop (%) |
> | :--- | :--- | :--- |
> | **Full Model** | 0.503 | n.a. |
> | w/o Hard Negative Sampling | 0.4197 | -16.5% |
> | w/o Rule Encoder (GRU) | 0.428 | -14.9% |
> | w/o Relation Embedding | 0.435 | -13.5% |
> | w/o Subgraph Encoder (GNN) | 0.455 | -9.5% |
>
> The results show that the Hard Negative Sampling component and the Rule Encoder (GRU) component are the most foundational ones, while the Subgraph Encoder (GNN) provides the critical “contextual lift” that pushes performance beyond static baselines. This result will be included in our revised manuscript, which is allowed to be submitted by December 3.

---

> ### Author Response · Authors · 2025-11-26
> **Response to Comments from Reviewer fMyx (Weakness 2)**
>
> 3. Efficiency Analysis (Weakness 2)
>
> Response: We thank the reviewer for the suggestions to improve the training cost. We note that training time is dominated by negative sampling.
> We can reduce training costs by sacrificing a little accuracy. We compared the default setting ($k_{pos}=5$, $k_{neg}=20$) against an efficient setting ($k_{pos}=1$, $k_{neg}=10$).  The results show we can achieve a **3.5x** speedup with negligible performance loss on most datasets.
>
> | Dataset | MRR (default setting) | MRR (efficient setting) | Drop (%) | Speedup |
> | :--- | :---: | :---: | :---: | :---: |
> | **WN18RR** | 0.4871 | 0.4849 | -0.45% | **3.5x** |
> | **FB15k-237** | 0.2796 | 0.2760 | -1.29% | **3.6x** |
> | **YAGO3-10** | 0.5031 | 0.4865 | -3.30% | **1.3x** |
>
> In the future, we will conduct more training optimization as suggested by the reviewer.

---

> ### Author Response · Authors · 2025-11-26
> **Response to Comments from Reviewer fMyx (Question 1)**
>
> 4. Subgraph Size vs. Rule Length (Question 1)
>
> Response: We found that this interaction depends heavily on graph sparsity.
> For super sparse graphs (WN18RR), deep reasoning is required through the use of a larger $L$ (e.g., $L=5$). Consequently, we must increase the subgraph depth ($k$) to capture the context of these distant nodes. Our ablation study on WN18RR confirms that performance (MRR value) improves steadily with depth:
> - $k=1$: 0.4327
> - $k=2$: 0.4807
> - $k=3$: 0.4824
> - $k=4$: 0.4871
>
> For denser Graphs like FB15k-237 and YAGO3-10, connections are local and abundant. Shorter rules ($L=3$) could be sufficient. Increasing $k$ beyond $1$ introduces exponential computational costs. Thus, we utilize $k=1$ for these datasets. We will update the Fig 3b with correct scale.

---

> ### Author Response · Authors · 2025-11-26
> **Response to Comments from Reviewer fMyx (Weakness 5)**
>
> 5. Related Works (Weakness 5)
>
> Response: We thank the reviewer for the references to IterE (WWW 2019) and RPJE (AAAI 2020). We will add discussions about these neuro-symbolic baselines into the revision.
>
> From a problem statement perspective, we acknowledge that these two methods can be used as baselines. However, both of them combined a rule-based and an embedding-based approach. It may not be a fair comparison with this group of methods.
> For IterE, the paper does not directly report the results on the datasets we experimented on. For RPJE, the only dataset they have in common with ours is FB15k-237, on which RPJE showed better results due to its use of both embedding and rules. For both approaches, given the response time window, we were not able to reproduce the results reported in their work using their published source code. To get reproducible results, we plan to conduct a more extensive comparison with them in the future.

---

> > ### Comment · Reviewer_fMyx · 2025-11-26
> >
> > Thanks for the authors' responses, they have addressed some of my concerns. Considering the responses from the authors and the opinions of other reviewers, I maintain the current positive score.

---

> > > ### Author Response · Authors · 2025-11-26
> > >
> > > We thank the reviewer for reading our responses and for your encouraging feedback!

---

### Author Response · Authors · 2025-12-03
**SUMMARY OF REBUTTAL**

Dear Area Chair,

We were sorry to learn of the OpenReview data leak and the resulting disruption to the conference. We understand this creates significant additional work for you, and we sincerely appreciate you stepping in to evaluate our submission under these challenging circumstances.

Because reviews and scores have been reverted to their pre-rebuttal state, we want to ensure you have our manuscript’s full context of the discussion period that occurred before the reset.

Status of reviewer scores (prior to revert). Before the system was reset, our rebuttal was well-received.
- Reviewer 4PdS **raised their score from 4 to 6**. The reviewer explicitly stated: “The author addressed most of my concerns, so I decided to raise my rating to 6: marginally above the acceptance threshold. But would not mind if paper is rejected”.
- Reviewer fMyx maintained a positive score of 6. After reviewing our new experiments, the reviewer stated: “Thanks for the authors’ responses, they have addressed some of my concerns. Considering the responses from the authors and the opinions of other reviewers, I maintain the current positive score.”
- Reviewer orxV maintained their score of 2. The reviewer noted a performance gap compared to NBFNet. We clarified that while NBFNet is a strong neural baseline, SLogic targets a different goal: learning interpretable logical rules for Knowledge Graph Completion (KGC). SLogic prioritizes generating explicit, human-readable logical rules, whereas NBFNet operates on neural representations of graph paths. Our case study demonstrates that human-readable rules and their query-specific rule scores offer explicit explanations for KGC process. SLogic outperforms all the other rule based methods on two out of the three benchmark datasets that we tested on, and ranks as the second-best on one dataset (FB15k-237).

Summary of revisions. We have uploaded a revised manuscript that clarifies our contribution, incorporates more discussions on related works, and includes the results generated during the rebuttal. To save you time, here is a summary of the key improvements.

1. We reframed the core contribution & novelty to address Reviewer orxV’s critique regarding the novelty of rule weighting. We clarified the manuscript’s positioning in the Abstract and Introduction.
- We revised the statement regarding the rules used in existing methods. We moved away from the claim that prior methods treat rules as universal or static. Instead, we now explicitly acknowledge that while existing neuro-symbolic methods learn confidence scores,
however they typically assign a fixed global weight to each rule.
- We provided additional explanations to highlight how SLogic differs from other rule-learning approaches. SLogic is now precisely defined as the first framework to learn query-dependent rule weights. We clarified that a rule like bornIn(X, Y ) ∧ locatedIn(Y, Z) → livesIn(X, Z) might have high confidence globally but low relevance for a specific query, an important distinction that only SLogic captures via its subgraph-informed scoring.

2. We expanded and restructured the Related Works section (Section 2) to accurately situate SLogic among the foundational methods identified by Reviewers orxV and fMyx.
- We added Markov Logic Networks (MLNs) and pLogicNet to the beginning of the rule-based section. We framed these as foundational works that combine logic with probabilistic models but distinguished them by noting their reliance on global schema weights rather than query-specific adaptation.
- We added discussion of the use of hybrid frameworks such as IterE and RPJE, as suggested by Reviewer fMyx. We described these as hybrid approaches that iteratively update embeddings and rules. We clarified that while they bridge the symbolic-subsymbolic gap, they still operate on the premise of global rule validity, contrasting them with SLogic’s local re-ranking mechanism.
- We clarified the difference between our work and other neural baselines such as NBFNet to accomodate Reviewer orxV’s request for discussion/comparison against state-of-the-art GNNs. We acknowledged NBFNet’s effectiveness in capturing all-path information but distinguished SLogic’s role based on symbolic interpretability. While NBFNet operates on neural representations to score paths, SLogic prioritizes generating explicit, human readable symbolic rules. This positions SLogic as a distinct approach for applications where explicit logical justifications are required.

3. We clarified and contextualized our use of subgraphs and subgraph features in Section 4.1, addressing reviewer orxV’s concern that they constitute heuristic engineering. We cited seminal works SEAL and GraIL in the framework section. We used these references to establish that extracting enclosing subgraphs and applying relative node labeling is a proven, standard inductive bias for learning structural reasoning patterns.

---

> ### Author Response · Authors · 2025-12-03
> **SUMMARY OF REBUTTAL (Cont.)**
>
> 4. We strengthened the empirical results & their robustness (Section 5.1). (i) We improved the method’s performance on FB15k-237 dataset. By analyzing rule coverage, we identified that dense graphs create rules that leads to too many entities, thus harder to be differentiated. We introduced a rule coverage penalty (λ), which raised the MRR on FB15k-237 from 0.28 to 0.30. This result makes SLogic highly competitive, ranking as the second-best rule-based method on FB15k-237 dataset, just behind RLogic. (ii) We verified the robustness of our results. To ensure reliability (problem raised by reviewer 4PdS), we repeated experiments across 5 random seeds. We reported the standard deviations (σ ≈ 0.0007), confirming that SLogic’s performance gains are statistically robust and not due to initialization noise.
>
> 5. We added new experimental results for ablation study and efficiency analysis (Sections 5.3 and 5.5).
> - We added a study showing the impact of query-dependent scoring mechanism. We added a direct comparison showing that SLogic’s dynamic scoring yields a +17.6% to +25.4% performance gain over a static global baseline, quantitatively proving the value of context-aware re-ranking.
> - We added an ablation study of model components to address a comment by Reviewer fMyx. We added a full ablation study demonstrating that Hard Negative Sampling, the Rule Encoder (GRU), and the Subgraph Encoder (GNN) are all critical, with removals causing performance drops of 9.5% to 16.5%.
> - We added more results to analyze training efficiency. We demonstrated an efficient training configuration ($k_{pos} = 1, k_{neg} = 10$) that achieves a 3.5x speedup on WN18RR and FB15k-237 with negligible performance loss (< 1.3%), offering a practical trade-off for resource constrained scenarios.
>
> 6. We have revised the manuscript to transparently acknowledge the limitations noted by Reviewer orxV regarding rule scope (Section 6). We added a dedicated Discussions subsection explicitly stating that SLogic currently focuses on chain-like Horn clauses. While we capture some tree-like patterns via inverse relations, we list the extension to complex non-chain structures, the reduction of computational overhead, and the exploration of better graph sampling strategies as clear directions for future work.
>
> We hope this summary assists you in your evaluation.
>
>
> Best regards,
>
> The Authors

---

### Meta-Review · Area_Chair_wKJH · 2026-01-07

**Summary:**

Two reviewers gave the paper a positive assessment (score 6). However, One reviewer raised concerns about inconsistent performance on FB15k-237, high training cost, missing ablation studies, and omission of key related works. He/She assigned a low score (2), criticizing the work as heavily reliant on heuristic engineering.

**Reviewer Concerns:**

The authors addressed nearly all technical points raised.

**Reviewer Scores:**

Reviewer fMyx (initial 6) explicitly stated they maintain their positive score after rebuttal; final score: 6.
Reviewer 4PdS (initial 4) explicitly raised their rating to 6 post-rebuttal; final score: 6.
Reviewer orxV (initial 2) reaffirmed their original rating and did not indicate any change; final score: 2.

---

### Decision · Program_Chairs · 2026-01-26

Reject